# Management Optimization of Electricity System with Sustainability Enhancement

**Wei Hou** [1,2,*], **Rita Yi Man Li** [3] **and Thanawan Sittihai** [1]

1   Chakrabongse Bhuvanarth International Institute for Interdisciplinary Studies, Rajamangala University of Technology Tawan-Ok, Bangkok 10400, Thailand; thanawansittihai@gmai.com
2   State Grid Beijing Electric Power Corporation, Beijing 100031, China
3   Sustainable Real Estate Research Center, Hong Kong Shue Yan University, Hong Kong, China; ritayimanli@gmai.com
*   Correspondence: hhbbj1108@163.com

**Abstract:** Based on new policies and social changes, renewable energies have highly penetrated electrical systems, making the system more vulnerable than before. On the other hand, it leads to congestion and competition within the network. To this end, this paper developed a probabilistic multi-objective-based congestion management approach and applied it to the optimal transmission switching (OTS) strategies, to maximize system suitability and minimize total production costs. A point estimation economic method (PEM) has been applied, as one of the best management and economic tools to handle the uncertainties associated with a wind turbine's power production and load demand (LD). Results demonstrate the effectiveness and merit of the proposed technique, compared to the existing one, which can lead to higher reliability and sustainability for the grids.

**Keywords:** dynamic thermal rating; probabilistic energy demand; two points approximation scheme; wind power; congestion management; electricity optimization; MPSO multi-objective optimization; suitability

## 1. Introduction

Electric utilities must use renewable energy resources, due to the growing demand for power and the increase in the world population. The popularity of such resources was, also, boosted by growing environmental concerns and reducing investment prices. Renewable energy resources, such as wind, are one of the most widely used resources, with 651 GW of installed capacity globally, in 2019 [1]. A significant challenge for network operators is associated with wind power fluctuations [2]. Wind power resources present a number of challenges in the transmission networks (TNs) operation, including that many lines lack sufficient capacity for transmitting the energy, so dealing in the day-ahead power market leads to congestion [3]. Generally, congestion of transmission happens whenever variations in demand/production cause a transfer of power that reaches or exceeds the transmission network's physical capacity. Additionally, the transmission infrastructure on the system is rapidly deteriorating. Consequently, the amount of congestion in the grid enhances markedly, which is a major factor in the increase in power production costs and the restriction of the use of renewable energy [4]. It is difficult and costly to construct new transmission lines (TLs) to reduce congestion, so utilities are seeking a more cost-efficient technology to make use of the current infrastructure [5]. The flexible AC-TS (FACTS), high voltage DC system, and enhanced conductor system are developing transmission technologies, which are increasing the capacity of the transmission network. Nevertheless, a number of technologies, such as FACTS controllers and large-scale energy storage, require significant investment. Accordingly, novel transmission technologies, designed to utilize the potential of the current transmission infrastructure, would be more appealing and are becoming, increasingly, used in practice. As a result, transmission service providers are

searching for alternatives that could be used effectively for their networks. This paper uses DTR and network topology optimization (NTO) as well as, for improving the network's reliability, increasing wind power's penetration rate and reducing congestions of TL.

A network topology optimization method can be used to take advantage of the current transmission infrastructure, to achieve a timely and significant purpose: The transmission network's performance and flexibility can be improved. As early as the 1980s, the concept of changing transmission network topology was offered [6], giving system operators the chance to temporarily remove TLs from service, giving them the flexibility they need to manage transmission network topologies. As a precaution or corrective measure, the TL is turned on/off to decrease voltage violations, line overloads, secure the system, and restore the load when an outage has occurred [7]. Transmission switching is also examined, as a means of harnessing the flexibility of the current transmission infrastructure to reduce the operational costs of the system, in addition to being a control action or as a solution to reduce system losses. Optimal transmission switching (OTS) was, firstly, introduced in [8], as part of the DC optimal power flow. A binary variable shows the on/off status of TLs, within the suggested formulation, while [9] extended the OTS real-time utilization to the AC optimal power flow (ACOPF) context. The two-level frequent structure was presented, using $2^{th}$-order cone programming, for generating ideal switching solutions at high levels; afterward, those solutions were screened for achieving AC reliability at low levels. The transmission switching of power systems is proven to lower the investment cost.

Through the development of heuristic solution techniques, transmission switching operations have become more adaptable, by finding solutions to OTS problems more quickly. Stochastic optimal transmission switching (SOTS) requires a suitable stochastic programming formulation. The SOTS must, explicitly, represent and consider the uncertain renewable generation and loads, while determining the OTS decisions. Then, [10] recommended an OTS model with adaptive robust optimization (RO), according to the uncertainty of net LD, and [11] used a linearized OTS model, according to AC OPF, for adapting to the random nature and intermittent nature of wind power for practical applications. Finally, [12] offered a very conservative TL switching solution, using an RO that just took into account the worst-case uncertainty.

Along with optimizing stochastic and robust programming, probable power flow has proven effective at examining the uncertain nature of electric power systems as well, and is used widely. Point estimation methods (PEM) are adapted, by such methods, to a wide range of applications [13]. Generally, Static line ratings (SLRs) are used to plan and operate many TLs. Severe weather conditions limit TS ratings, in this case. It might be difficult to fully utilize the TLs, due to this conservative method [14]. Therefore, transmission service providers are exploring alternatives, in order to optimize the use of the transmission network. As an example, one of those methods uses the DTR of TLs, which is dependent on the actual climate, for creating conditions of actual loading [15]. Taking the benefit of the cold temperatures and high winds that occur during such events will enable grid operators to significantly increase the thermal limit of TLs with DTR [15]. TS operators benefit most from this solution, as by utilizing current assets for TLs, capital investment in this part of the system is much less, in comparison to reconstructing more resilient systems from scratch. Power systems have been the subject of many DTR study programs. DTR's integration into real-time operations improved congestion management (CM) as well as related costs throughout such alterations and events in [16]. Moreover, Ref. [17] discussed the aging impact of transmission overhead lines based on the DTR technique, and its parametric uncertainty towards line failure probability. Despite this, the majority of research exists on the basis of fixed networks. Coordinating DTR and other new transmission-control technologies should be investigated further. The OTS and DTR share a similar benefit, namely that, together, they can add more power over the transmission network, by mitigating system limitations and decreasing operational costs. In summary, the co-optimization of these two technologies in the power system would increase its efficiency effectively. Power system reliability has been enhanced recently, by

combining DTR and OTS with other technology. In addition, Ref. [18] discussed DTR progress scheduling with NTO deployment and its reliability evaluation, while Ref. [19] suggested combining OTS and DTR in a unit commitment problem, to increase power distribution uniformity. Moreover, Ref. [20] discussed the effects of OTS and dynamic thermal ratings on decreasing emission of carbon dioxide, particularly in the case of diversifying energy resources. A key aim of this research is to schedule the power system day-ahead, by simultaneously implementing OTS and DTR. Compared to the previous investigations, there are still a limited number of suitable limitations, which are, directly, incorporated into the formulation of OTS problems, for assuring that an OTS problem is truly networked, and that the load and wind power are random and variable.

The probable multi-objective CM, according to the PEM using the DTR and OTS methods, is presented in the present study, which supports utilities in making better use of current transmission infrastructure, by providing better congestion relief efficiencies. In total, two levels of the problem are planned. Firstly, MPSO examines choices from a range of possible switching schemes for TLs, and, then, it sends the results to the next level. In the next step, (2PEM + 1) has been applied, for solving the probable optimum power flow problems that have been utilized to estimate the expected costs of production and the reliability index, related to various strategies.

The paper consists of the following parts: Part 1 provides an overview of the study. Part 2 deals with the modeling of system uncertainties. Part 3 introduces the probabilistic power flow (PPF) estimation process (2PEM + 1). Part 4 discusses the problem formulation. Part 5 presents the MPSO algorithm, whereas Part 6 covers the test system and scenarios. Part 7 concludes the paper.

## 2. Modeling of Uncertainties

The increasing uncertainty associated with load and renewable energy sources is one of the major characteristics of modern power grids. There are various methods to handling uncertainty in the electrical grid, and they usually have been grouped into RO methods, probable methods, and interval-set analyses. RO models uncertainties in the form of intervals and uses optimization for solving state boundary issues. Interval set decomposition involves demonstrating the uncertainty in the form of a set/interval and using an interval set method for estimating the output boundary. In probabilistic models, uncertain parameters are modeled as random variables, with known probability density functions, and propagations of uncertainty are modeled through analytic methods or Monte Carlo simulation. The present paper uses the probabilistic method, since this can be a widely applied method, in order to handle uncertainties in electrical grids [1] (See Appendix A).

Regarding the distribution types, please note it that it is widely accepted in the literature, [1,21], that normal distribution and the Weibull type can be used for modeling the behavior of the load and wind speed/power. Nevertheless, it should be noted that any other PDF function can be used in the same manner, without loss of generality. In other words, we can use other PDF types in quite the same way as we used the normal or Weibull functions. Meanwhile since these assumptions are made based on long-term big data, there is no way to check their accuracy in this work. In fact, in practice, we need to, first, make an initial data analysis to find the most-fitting PDF, based on our real-time data, and, then, start working with our model.

### 2.1. Modeling of Probable Load

There are two parameters that define probability distribution functions (PDFs); the mean ($\mu$) and standard deviation ($\sigma$) of the uncertain parameter are described in the following way [1]:

$$\text{PDF(s)} = \frac{1}{\sigma\sqrt{2\pi}} * exp^{\left(\frac{(-(S-\mu)^2)}{2\pi^2}\right)} \tag{1}$$

Here, the apparent power of the load is shown by $S$, and the assumption is that every bus has a mean $(\mu)$ that equals the base load and a standard deviation $(\sigma)$ of $\pm 5\%$ of the base load [1].

### 2.2. Probabilistic Wind Modeling

Wind turbines (WTs) generate power based on the wind speed $(v)$, which can typically be modeled using the Weibull distribution PDF [22]:

$$\text{PDE(v)} = \left(\frac{k}{c}\right)\left(\frac{v}{c}\right)^{k-1} exp\left[-\left(\frac{v}{c}\right)^k\right] \tag{2}$$

Here, the scale factor is shown by $c$, and the shape factor of the Weibull function is indicated by $k$.

The PDF of the wind speed can be supposed to be known in every region, and the variation of wind speed to WT output power can be determined via [23]:

$$P(v) = \begin{cases} 0, & v \geq v_o \quad or \quad v \leq v_i \\ P_r\left(\frac{v-v_i}{v_r-v_o}\right) & v_i \leq v \leq v_r \\ P_r & v_i \leq v \leq v_r \end{cases} \tag{3}$$

Here, $v_i$ indicates the cut-in wind speed, $v_r$ shows the rated wind velocity, $v_o$ shows the cut-out wind speed, and $P_r$ represents the rated power.

### 2.3. Probabilistic Line Rating Modeling

Various approaches exist for implementing DTR, always requiring various input parameters. Those approaches are: [24] (1) predicting DTR based on climate forecasts and system loading; (2) an estimate of DTR based on indirect measurements; and (3) an actual DTR assessment based on real climatic information, as applied in this study. Therefore, the ampacity of the line must be considered as a probabilistic variable, in which the PDF of a thermal limit (MVA) would correspond to a generalized extreme value distribution, as illustrated in Equation (4). [1,21].

$$\text{PDF(t)} = \left(\frac{1}{\sigma_1}\right)\cdot\left(1+\xi\frac{(t-\mu l)^{\frac{-1}{\xi}}}{\sigma_1}\right)\cdot e^{\left(-\left(1+\xi\frac{(t-\mu l)^{\frac{1}{\xi}}}{\sigma_1}\right)\right)} \tag{4}$$

Here, the DTR of the line is shown by $t$, the location parameter is represented by $\mu_l$, the shape parameter is shown by $\xi$, and the scale parameter is represented by $\sigma_l$. Weather conditions determine these parameters, which are computed according to [25]. Therefore, the DTR mean has been achieved for every period (for example, season) and is applied as a limit to the formulas of power flow [1].

## 3. Probabilistic Power Flow

This paper has proposed a stochastic framework based on PEM, to handle the uncertainties of the problem, including the renewable sources' output power uncertainties and the load demand uncertainties. Therefore, Section 2 is devoted to describing this section completely. The core idea in PEM is to replace the PDF functions with some appropriate fitting concentration points. In this approach, the PEM solves probabilistic problems via a deterministic process, although it needs less computation [13]. Therefore, the PEM would decompose the stochastic problem into a 2m + 1 equivalent deterministic problem, with different probabilities. PEMs have the advantage of requiring basic information about random variables, in order to model them effectively. The skewness, variance, and kurtosis of the variables are included in the data. This study uses the (2m + 1) scheme, in which the kurtosis of the input random parameters is taken into account, and just one more computation has been performed [26]. More details, including the complete formulations,

are provided in this paper, which are highlighted in yellow. The PPF problem is solved using the scheme (2m + 1), in the following way [1]:

$$\xi_{l,k} = \frac{\lambda_{l,3}}{2} + (-1)^{3-k}\sqrt{\lambda_{l,4} - \frac{3}{4}\lambda^2_{l,3}}$$
$$k = 1,2, \ \xi_{l,3} = 0 \tag{5}$$

$$p_{l,k} = \mu_{pl} + \sigma_{pl}k = 1,2,3 \tag{6}$$

$$w_{l,k} = \frac{(-1)^{3-k}}{m\xi_{l,k}(\xi_{l,1} - \xi_{l,2})} \quad k = 1,2 \tag{7}$$

$$w_{l,3} = \frac{1}{m} - \frac{1}{m(\lambda_{l,4} - \lambda^2_{l,3})} \tag{8}$$

Here, $l, k$ represents the standard location, $\mu_{P1}$ shows the mean, $\sigma_{Pl}$ indicates the standard deviation, and $\lambda_{l,j}$ shows the $j$th standard central moment of the input random parameters $p_l$. Based on Equation (11), location $l, 3 = 0$ gives $p_{l,3} = \mu_{pl}$ and, thus, the locations are the same $\left(\mu_{p1}, \mu_{p2}, \ldots, \mu_{pl}, \ldots, \mu_{pm}\right)$ point. Therefore, it should be possible to perform a single calculation at this location, given equivalent weight $w_0$, in the following way:

$$w_0 = 1 - \sum_{l=1}^{m} \frac{1}{m(\lambda_{l,4} - \lambda^2_{l,3})} \tag{9}$$

In addition, Equation (11) proves that the scheme yields non-real locations if $\lambda_{l,4} - \frac{3}{4}\lambda^2_{l,3}$ has a negative value. Furthermore, in power system problems, the probability distributions have been typically applied to the normal, uniform, and binomial models; thus, the places can always be actual.

The PPF issue, utilizing the (2m + 1) layout, is solved by modeling the power flow input information as random variables, and calculating the weights and places by applying Equations (11) and (13). A solution can be found in [26]:

$$E\left(Z^j\right) = \sum_{l=1}^{m}\sum_{k=1}^{2} w_{l,k}(Z(l,k))^j + w_o Z_o^j \tag{10}$$

$$(l,\ k) = F\left(\mu_{p1},\ \mu_{p2}, \ldots, P_{1,k}, \ldots, \mu_{pm}\right) \ , \ k = 1,2 \tag{11}$$

$$Z_o = F\left(\mu_{p1},\ \mu_{p2}, \ldots, \ldots, \mu_{pm}\right) \tag{12}$$

Here, $Z(l, k)$ shows the output of the RVs associated with the $k$th concentration $(\mu_{p1}, \mu_{p2}, \ldots, P_{1,k}, \ldots, \mu_{pm})$ of random parameters, representing the relation between the output and input in the probable power flow (PPF). The gathering scheme determines the dense number of definite PFs needed to be executed. $Z(l,\ k)$ has been used for assessing the raw moments of the yield, and the computation ends once entire centralizations of data RVs have been taken into account. Afterwards, applying Equation (12), the analyzed raw moments of the yield have been used for calculating the necessary statistical data [1].

## 4. Formulation of the Issue

### 4.1. Objective Functions

In TNs, the TL switching method reduces congestion. Through optimization, TLs that need to be disconnected would be identified, decreasing congestion. In order to determine the optimal switching plan, in terms of maximum probabilistic reliability and lowest cost of production, a multi-objective-based methodology is proposed in this study. Following are details on the objective functions.

### 4.1.1. Overall Cost of Power Production

In order to realize this goal, total network savings must be maximized, via a reduction in total power production costs, which are dependent on physical limitations, such as TL flow limitations and voltage of bus restrictions. It should be noted that it is well perceived that the cost function is considered in a quadratic format, since the case study is the transmission system. From a technical point of view, this models the nonlinear opening and closing process of the steam valves, which looks like sinusoidal curves. It is clear that we have to use a linear equation (rather than a quadratic function), if the case study is a distribution system. The paper uses the secondary supply bid price function:

$$min \sum_{i=1}^{n_g} \left( a_i + b_i P_{gi} + c_i P^2{}_{gi} \right) \tag{13}$$

Subject to:

$$P_{gi} + P_{wi} + P_{Di} - V_i \sum_{j=1}^{N_b} V_j \left( G_{ij} \cos \delta_{ij} + B_{ij} \sin \delta_{ij} \right) = 0 , \ i \in N_b \tag{14}$$

$$Q_{gi} + Q_{wi} + Q_{Di} - V \sum_{j=1}^{N_b} V_{ji} \left( G_{ij} \cos \delta_{ij} - B_{ij} \cos \delta_{ij} \right) = 0 , \ i \in N_b \tag{15}$$

$$P_{gi}^{min} \leq E\left( P_{gi} \right) \leq P_{gi}^{max} , i \in N_G \tag{16}$$

$$Q_{gi}^{min} \leq E\left( Q_{gi} \right) \leq Q_{gi}^{max} , i \in N_G \tag{17}$$

$$V_{gi}^{min} \leq E\left( V_{gi} \right) \leq V_{gi}^{max} , i \in N_G \tag{18}$$

$$V_{Li}^{min} \leq E(V_{Li}) \leq V_{Li}^{max} , i \in N_L \tag{19}$$

$$\left( S_{li} \right) \leq S_l^{max} * \gamma_k \ \ , i \in N_L \tag{20}$$

$$\delta_i^{min} \leq \delta_i \leq \delta_i^{max} , i \in N_b \tag{21}$$

$$\sum_{k=1}^{N_L} (1 - \gamma_k) \leq \varphi \tag{22}$$

Here, $Q_{gi}$ and $P_{gi}$ represent the reactive and active power production of the $i$ generator unit; $Q_{wi}$ and $P_{wi}$ show the reactive and active power generation of wind farm $i$; $Q_{D1}$ and $P_{D1}$ represent the reactive and active LD at load bus $i$; $P_{gi}^{max}$, $P_{gi}^{min}$, $Q_{gi}^{max}$, and $Q_{gi}^{min}$ show the maximum and minimum restrictions of the reactive and active power injection of the $i$th generator agent; $V_{gi}^{min}$ and $V_{gi}^{max}$ represent the minimum and maximum limits of the voltage magnitude at bus $i$; $V_j \angle \delta_j$ and $V_i \angle \delta_i$ show the termination buses $j$ and $i$ voltages; $S_l$ represents the power flow via the line, and $S_l^{max}$ shows the loading limit; $B_{ij}$ and $G_{ij}$ represent the substance and conductance of the branch linked between bus $i$ and $j$; $\varphi$, $N_b$, $N_g$, and $N_L$ indicate the groups of the switching lines number, branches number, generator buses, and buses of load, respectively; $\gamma_k$ shows the state of line $l$ that can be 0 or 1 (open/close); and $a_i$, $b_i$, and $c_i$ represent generator fuel ratios.

### 4.1.2. Probable Reliability Objective

Maximum probabilistic reliability has been regarded as an additional target to optimize, in the suggested multi-objective optimization problem. OTS strategy can effectively change the structure of the grid and, thus, provide new power flow paths, through which the optimal values of the objective functions would enhance. OTS would, then, minimize the power losses and cost, enhancing the EENS in the system. From the market point of view, this is very useful and can be considered as a powerful ancillary service for the market players. The expected energy not supplied (EENS) factor has been applied as the OTS's

goal of OTS, showing the reliability and efficiency of the grid, in the status of structure reconfiguration. The probabilistic analysis state enumeration method is used to calculate the EENS factor of the TS, which can follow each OTS plan [27]. The objective function, which should be minimized, can be described via Equation (23):

$$min(EENS) = \sum_{s \in \Omega} P_t(s) \cdot PC(s) \tag{23}$$

$$P_t(s) = \left( \frac{\lambda_i}{\lambda_i + \mu_i} \right) \prod_{j=1}^{\psi} \left( 1 - \frac{\lambda_i}{\lambda_i + \mu_i} \right) \tag{24}$$

Here, $PC(s)$ is the total load curtailment and $P_t(s)$ is the probability of occurrence of grid status $s$; $\Omega$ shows the set for grid status $s$; $\lambda_i$ and $\mu_i$ represent the blackout and element $i$ maintenance rate; and the accessible parts number is shown by $\psi$.

Failures of TL are modeled as an independent single blackout $(N-1)$; therefore, just first-order contingencies are taken into account, and, as far as is related to the cause, that will not be affected by any concurrent failures. Reconstruction state probabilities have been computed in Equation (24), in which the probability of occurrence of every contingency (s) has been obtained, by multiplying the existing part's probability by the failed part's probability. The overall load curtailment, $PC(s)$, of any status $s$, has been computed via an ACOPF method, as illustrated in the Equations (25)–(29) sub-problem, which can minimize the overall load decrement of every possibility:

$$min \sum_{i \in N_b} PC_i \tag{25}$$

Subject to:

$$P_{gi} + P_{wi} - P_{Di} + PC_i - V_i \sum_{j=1}^{N_b} V_j \left( G_{ij} \cos \delta_{ij} + B_{ij} \sin \delta_{ij} \right) = 0 , \ i \in N_b \tag{26}$$

$$Q_{gi} + Q_{wi} - Q_{Di} + QC_i - V_i \sum_{j=1}^{N_b} V_j \left( G_{ij} \sin \delta_{ij} - B_{ij} \cos \delta_{ij} \right) = 0 , \ i \in N_b \tag{27}$$

$$0 \leq PC_i \leq P_{Di} i \in N_b \tag{28}$$

$$0 \leq QC_i \leq Q_{Di} i \in N_b \tag{29}$$

It is necessary to determine the reactive and active power at the bus of the generator and the magnitude of the voltage of bus in the inequality restriction, according to Equations (16)–(20).

The EENS index was assessed in networks using the (2PEM + 1) process, and the load's uncertainties and WT's power have been taken into account. Figure 1 shows the steps for calculating the EENS index, using the PEM-based approach.

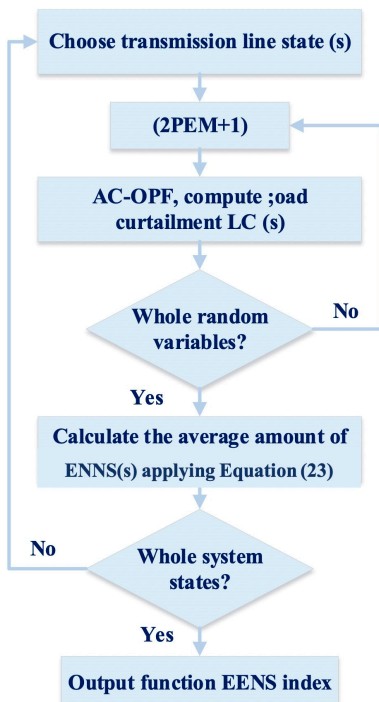

**Figure 1.** Process for determining the EENS reliability index.

## 5. Solution Method

### 5.1. Enhanced Particle Swarm Optimization

PSO is an evolutionary optimization technique, in order to minimize an objective that mimics the behavior of flocks of birds flying overhead or a group of fish. Particle swarm optimizers are comprised of particles and update empirical data about a search space, iteratively. Individuals in the population illustrate potential solutions to problems and can be viewed as particles that move in a $\psi$-dimensional search space.

Particles in a general PSO algorithm will adjust their position in accordance with their experiences and those of their neighbors, such as their current position, velocity, and the best prior position. By including the worst experience of each particle, MPSO enhances convergent performance of PSO and offers additional exploration capacity to the swarm [28]. By remembering its worst experience, the particle explores the search space more efficiently, to determine the best solution area. The MPSO algorithm updates positions and velocities of particles, in the following way [28]:

$$
\begin{aligned}
v_i^d(k+1) = \xi v_i^d(k) &+ c_1 \times r_1 \times (P^d{}_{best,i} - X_i^d(k) + c_2 \times r_2 \times (G^d{}_{best} \\
&- X_i^d(k)) + c_3 \times r_3 \times \left( X_i^d(k) - P^d{}_{worst,i} \right)
\end{aligned}
\tag{30}
$$

$$
x_i^d(k+1) = x_i^d(k) + \sigma v_i^d(k+1)
\tag{31}
$$

Here, the current velocity of the $i$th particle is shown by $v_i^d(k)$, $i = 1, \ldots, P$, where $P$ shows the population size; $k$ indicates the kth iteration; superscript $d = 1, \ldots, \psi$ shows the dimensions of the particle; $P_{best,i}$ represents the best prior location of the $i^{th}$ particle; $P_{worst,i}$ shows the worst prior location of the $i$th particle; $G_{best}$ represents the best prior location between whole of particles in the swarm; $X_i^d(k)$ shows the current location of the $i$th particle; $c_1$, $c_2$, and $c_3$ represent the acceleration factors; $r_1$, $r_2$, and $r_3$ show the monotonous random numbers among 0 to 1; and $\sigma$ shows the learning factor. By using designed inertia weight $\zeta$, the previous updated feature is copied to the next iteration. When the greater $\zeta$ has been chosen, the previous $v_i^d(k)$ has a significant influence on $v_i^d(k+1)$. It should be noted that the PSO method in Equation (30) does not include the last section on the right part. The amount of MPSO variables have been presented in Table 1.

**Table 1.** Amount of MPSO variables.

| Variable | $P$ | $c_1$ | $c_2$ | $c_3$ | $\psi$ | $\sigma$ | $\zeta$ | $k_{max}$ |
|---|---|---|---|---|---|---|---|---|
| Value | 5 | 0.5 | 0.5 | 0.5 | 1 | $5 \times 10^{-3}$ | 1 | 30 |

### 5.2. Islanding Prevention

System-wide cascading interruptions and complete blackouts can be avoided, by using power system island detection. When the TL is disconnected, it can lead to bus isolation. Consequently, the optimization process must include measures to avoid islanding. The prior investigations have not included restrictions that could be directly incorporated into the OTS problem formulation, to make sure the network connection is properly enforced. A novel island detection process, which can provide a robust approach for connection matrix detection, is presented in this paper [29]. Figure 2 shows the suggested MPSO algorithm.

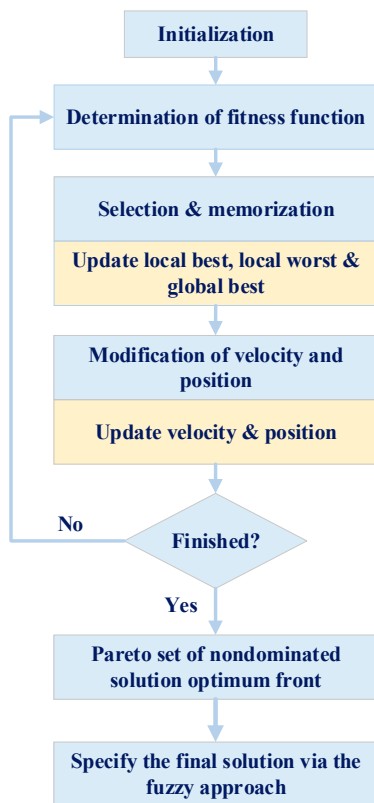

**Figure 2.** MPSO procedure.

### 5.3. Suggested Method

Two levels are planned for the complete problem. During the first level, MPSO determines the optimal TL switching strategies from various feasible ones, and the results are transferred to the 2th level. The 2th level involves the use of the (2PEM + 1), for the solution of the issues of the PPF, which can be necessary for determining the production cost and reliability index for different strategies.

## 6. Test System and Scenarios

An IEEE RTS 96 test system [30] is applied, for illustrating the suggested approach in cases where transmission switching is needed to reduce operating expenses and ease congestion. Interconnections among three identical 24-bus networks make this system possible. This system has an active power load of $8.55 \times 10^3$ (MW) and a reactive power of $1.74 \times 10^3$ (MVAr). A total of $9.832 \times 10^3$ (MW) active power and $2.001 \times 10^3$ (MVAr) reactive power have been added to the system. For every bus, the active and reactive power

has been expressed as the usual PDF, whose mean ($\mu$) equals the basic load and whose standard deviation ($\sigma$) equals 5% of the basic load.

Table 2 gives producing unit cost and kinds of data information. Figure 3 shows a schematic scheme of the RTS. Six wind farm generations, with 285 MW of installation capacity, have been permitted to be linked at bus 10, with 14 in every region of the RTS. There are 190 WTs in each wind farm. The whole optimization runs start with population size [$N = 100$] and the maximum number of iterations [$GN = 200$], as primary control parameters for the MO MPSO method. Pareto optimal fronts, with a maximum size of 25 solutions, were chosen. An analysis of the operational and economic advantages of implementing the OTS approach was conducted, using two scenarios.

**Table 2.** Producing agent kinds and cost data.

| Generator Unit | $U_{12}$ | $U_{20}$ | $U_{50}$ | $U_{76}$ | $U_{100}$ | $U_{155}$ | $U_{197}$ | $U_{350}$ | $U_{400}$ |
|---|---|---|---|---|---|---|---|---|---|
| Size (MW) | 12.00 | 20.00 | 50.00 | 76.00 | 100.00 | 155.00 | 197.00 | 350.00 | 400.00 |
| Fuel | Steam/Oil | CT/Oil | Hydro | Steam/Coal | Steam/Oil | Steam/Coal | Steam/Oil | Steam/Coal | Nuclear |
| **Fuel** ($/**Mbtu**) | 8.4 | 15.17 | 0 | 1.78 | 8.4 | 1.78 | 8.4 | 1.78 | 0.6 |
| **Cost** ($/**Mwh**) | 85.5 | 149.56 | 0.1 | 17 | 67.95 | 14.71 | 74.75 | 14.96 | 22 |

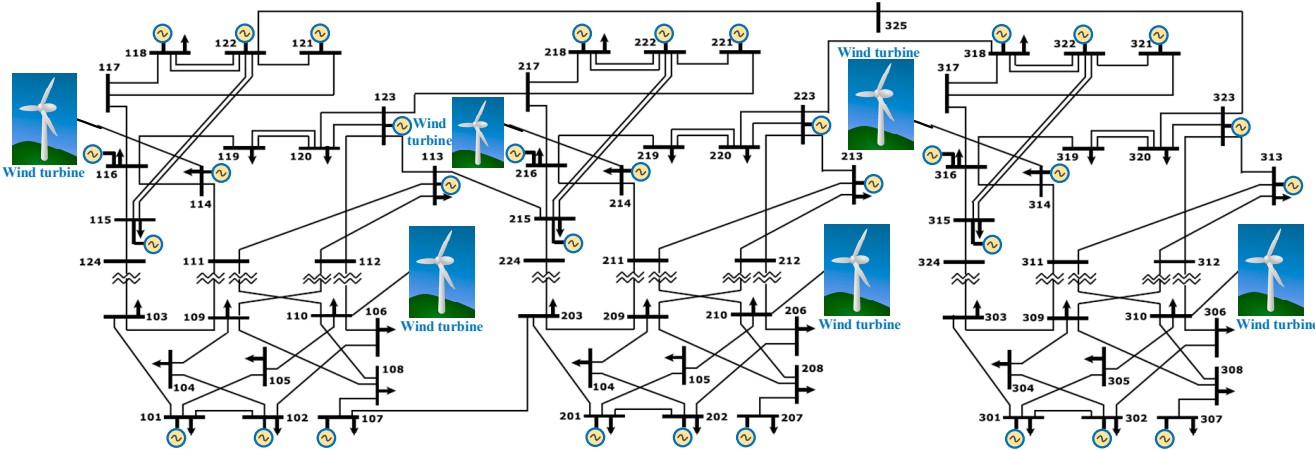

**Figure 3.** Enhanced IEEE RTS 96-bus network.

## 6.1. First Case Study: OTS with SLR

Figure 4 shows the Pareto optimal front for two objectives: probabilistic reliability and the production costs for DTR. MPSO has been used to derive the Pareto optimal front, and the fuzzy approach has been used for defining the BCS for diverse wind farm production capacities, according to Tables 3 and 4. In order to reduce grid dispatch prices with OTS, just one line is open ($\phi = 1$). For the optimization issue, the line that links bus 109 to bus 111 is switched off. Eight producing sets in the grid have their output power changed by this reconfiguration. According to Table 5, generator agents U179 and U100 decrease output, whereas generator agents U12, U350, and U67 increase output. Reducing the generation costs of generators, by transferring power from costly devices to low-price devices, without compromising the network's safety, is possible. Continuing the analysis for [$\phi = 2 \ to \ 4$], for every further open line, the cost of the network reduces, but at a decreasing rate. Various wind farm capacities have been investigated in four subcases. Tables 3 and 4 show the outcomes. As wind farm capacity increases, power production costs decrease, if the number of switched TLs is zero. Tables 3 and 4 show that if the number of TLs increases, the mean of the production cost of the network decreases, with an increase in the capacity of the wind farm, due to the greater dispatch of cheaper generators, according to Figure 5. As a consequence, if $N - 1$ security has been kept, *EENS* greatly enhances those probabilistic indicators for the four topologies, which have disconnected lines. Figure 6 indicates that up to four lines can be opened, without compromising the system operating constraints.

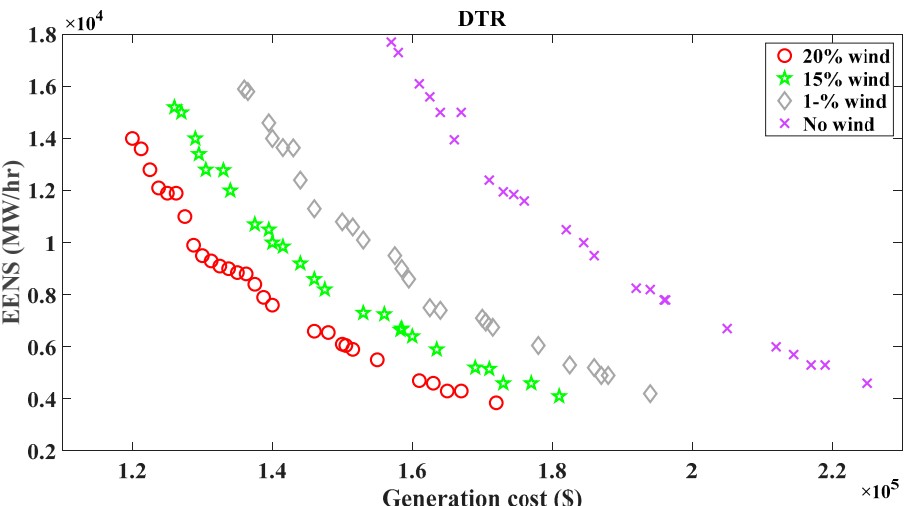

**Figure 4.** The EENS Pareto optimal front and overall production cost objective applying MOMPSO at ($\phi = 4$) for DTR.

**Table 3.** The BCS outcomes for OTS methods with SLR (with wind influence 0% and 10%).

| $\varphi$ | 0 | 1 | 2 | 3 | 4 |
|---|---|---|---|---|---|
| Wind penetration (0%) Lines open | - | [111–114] | [210,211]; [111–114] | [118–121]; [209–111]; [210,211] | [310,311]; [106–110]; [118–121]; [109–111] |
| $\mu[Gen.cost]$ ($/h) | 189,685 | 186,439 | 185,285 | 184,633 | 184,125 |
| $\mu[EENS]$ (Mwh/y) | $2.77 \times 10^3$ | $5.831 \times 10^3$ | $6.93 \times 10^3$ | $7.78 \times 10^3$ | $10.273 \times 10^3$ |
| Wind penetration (10%) Lines open | - | [111–114] | [210,211]; [109–111] | [110–112]; [109–111]; [215,216] | [312–314]; [106–110]; [118–121]; [109–111] |
| $\mu[Gen.cost]$ ($/h) | 164,860 | 161,340 | 160,215 | 159,730 | 159,310 |
| $\mu[EENS]$ (Mwh/y) | $2.49 \times 10^3$ | $5.31 \times 10^3$ | $6.33 \times 10^3$ | $7.08 \times 10^3$ | $9.273 \times 10^3$ |

**Table 4.** The BCS outcomes for OTS methods with SLR (with wind influence 15% and 20%).

| $\varphi$ | 0 | 1 | 2 | 3 | 4 |
|---|---|---|---|---|---|
| Wind penetration (15%) Lines open | - | [109–111] | [210,211]; [114–111] | [118–121]; [109–108]; [210–205] | [310–305]; [106–110]; [118–117]; [109–111] |
| $\mu[Gen.cost]$ ($/h) | 189,685 | 186,439 | 185,285 | 184,633 | 184,125 |
| $\mu[EENS]$ (Mwh/y) | $2.77 \times 10^3$ | $5.831 \times 10^3$ | $6.93 \times 10^3$ | $7.78 \times 10^3$ | $10.273 \times 10^3$ |
| Wind penetration (20%) Lines open | - | [119–114] | [210,211]; [109–103] | [118–121]; [109–108]; [210,211] | [311–314]; [106–110]; [218–222]; [109–104] |
| $\mu[Gen.cost]$ ($/h) | 164,820 | 143,620 | 142,510 | 140,715 | 140,030 |
| $\mu[EENS]$ (Mwh/y) | $2.36 \times 10^3$ | $5.01 \times 10^3$ | $5.95 \times 10^3$ | $6.69 \times 10^3$ | $8.83 \times 10^3$ |

**Table 5.** Alterations in generator output, following switching off the line (109–111).

| Generator Unit | $U_{12}$ | $U_{20}$ | $U_{50}$ | $U_{76}$ | $U_{100}$ | $U_{155}$ | $U_{197}$ | $U_{350}$ | $U_{400}$ |
|---|---|---|---|---|---|---|---|---|---|
| Size (MW) | 12.00 | 20.00 | 50.00 | 76.00 | 100.00 | 155.00 | 197.00 | 350.00 | 400.00 |
| Fuel kind | Oil/Steam | Oil/CT | Hydro | Coal/Steam | Oil/Steam | Coal/Steam | Oil/Steam | Coal/Steam | Nuclear |
| **Fuel** ($/**MBtu**) | 8.4 | 15.17 | 0 | 1.78 | 8.4 | 1.78 | 8.4 | 1.78 | 0.6 |
| **Cost** ($/**Mwh**) | 85.5 | 149.56 | 0.1 | 17 | 67.95 | 14.71 | 74.75 | 14.96 | 22 |
| Change in output (Mw) | 4.5 | 0 | 0 | 29.74 | −17.37 | 0 | −98 | 81.13 | 0 |

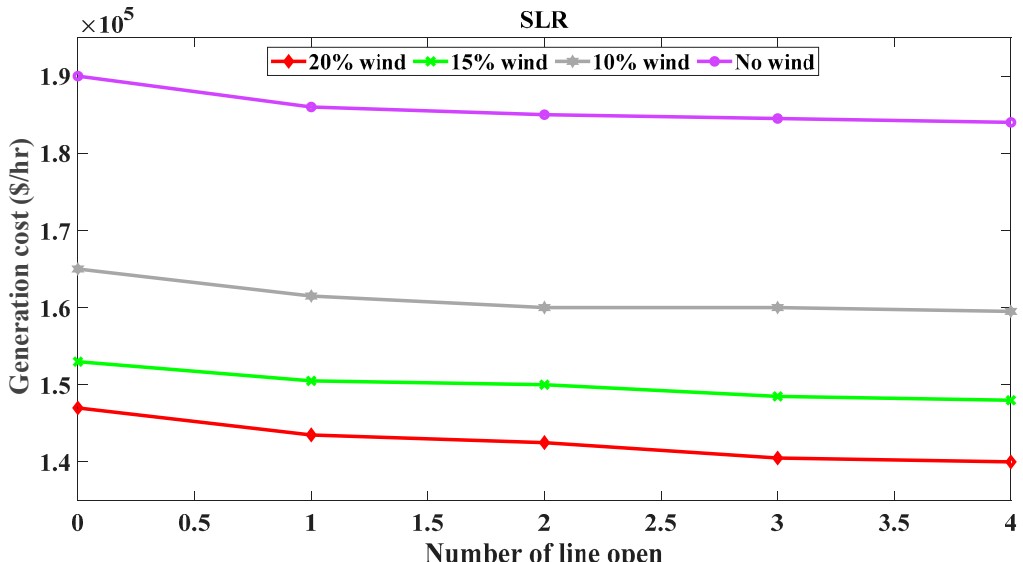

**Figure 5.** Anticipated producing cost for the OTS methods, for various wind levels for SLR.

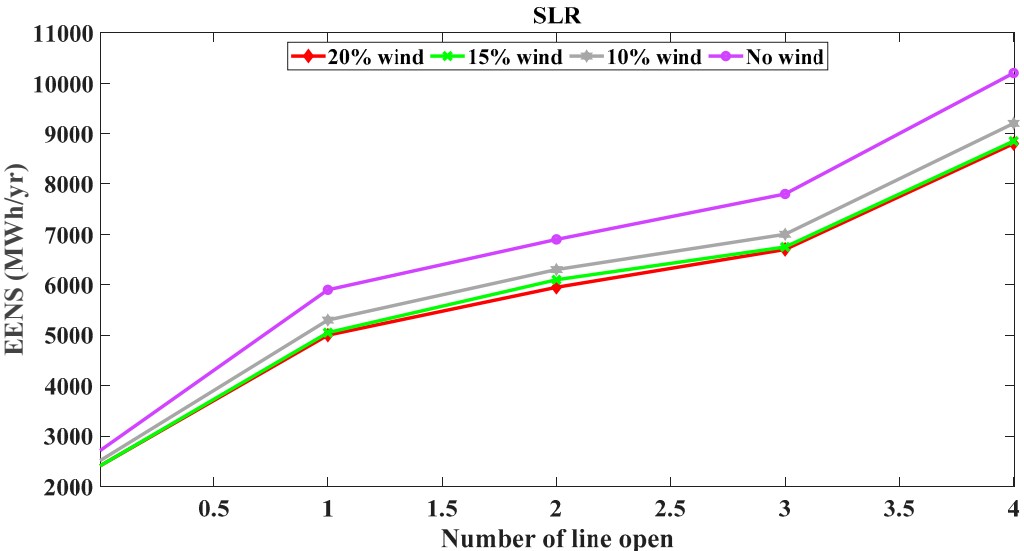

**Figure 6.** EENS for the OTS methods, for various SLR wind stages.

## 6.2. Case Study: OTS with DTR

Based on the suggested approach in Part 5, Tables 6 and 7 illustrate the achieved production cost and EENS index. The power, of the great-cost generator agent U197 in the DTR, has been decreased and moved to the small-cost generator agents U155, U76, and U100, compensating for the power loss of U197. Based on Figure 7, it can be concluded that up to four lines can be opened without compromising $N − 1$ security. The BCS achieved, for this case, shows that when opening TLs 310–311, 106–110, 118–121, and 109–111, the overall production reduces from $141,520$ USD/h to $131,830$ USD/h, as a result of the

greater dispatch of low-cost generators, and the system EENS of reliability relevant to the optimum OTS method changes from $2.1 \times 10^3$ MWh/yr to $7.53 \times 10^3$ MWh/yr, according to Figure 8. The dispatch cost-saving amount with OTS and DTR is 7.15%.

**Table 6.** The BCS outcomes for OTS methods with DTR (with wind influence 0% and 10%).

| $\varphi$ | 0 | 1 | 2 | 3 | 4 |
|---|---|---|---|---|---|
| Wind penetration (0%) Lines open | - | [109–111] | [210,211]; [109–111] | [118–121]; [109–111]; [210,211] | [310,311]; [106–110]; [118–121]; [109–111] |
| $\mu[Gen.cost]$ ($/h) | 189,625 | 186,412 | 185,235 | 184,587 | 184,098 |
| $\mu[EENS]$ (Mwh/y) | $2.56 \times 10^3$ | $5.42 \times 10^3$ | $6.44 \times 10^3$ | $7.23 \times 10^3$ | $9.53 \times 10^3$ |
| Wind penetration (10%) Lines open | - | [119–111] | [210,211]; [109–111] | [118–121]; [109–111]; [210,211] | [310,311]; [106–110]; [118–121]; [109–111] |
| $\mu[Gen.cost]$ ($/h) | 164,710 | 161,240 | 160,215 | 159,510 | 159,205 |
| $\mu[EENS]$ (Mwh/y) | $2.31 \times 10^3$ | $4.91 \times 10^3$ | $5.85 \times 10^3$ | $6.54 \times 10^3$ | $8.57 \times 10^3$ |

**Table 7.** The BCS outcomes for OTS methods with DTR (with wind influence 15% and 20%).

| $\varphi$ | 0 | 1 | 2 | 3 | 4 |
|---|---|---|---|---|---|
| Wind penetration (15%) Lines open | - | [109–111] | [210,211]; [114–111] | [118–121]; [109–108]; [210–205] | [310–305]; [106–110]; [118–117]; [109–111] |
| $\mu[Gen.cost]$ ($/h) | 153,101 | 150,05 | 149,520 | 148,490 | 147,850 |
| $\mu[EENS]$ (Mwh/y) | $2.41 \times 10^3$ | $5.01 \times 10^3$ | $6.12 \times 10^3$ | $6.77 \times 10^3$ | $8.93 \times 10^3$ |
| Wind penetration (20%) Lines open | - | [119–114] | [210,211]; [109–103] | [118–121]; [109–108]; [210,211] | [311–314]; [106–110]; [218–222]; [109–104] |
| $\mu[Gen.cost]$ ($/h) | 141,520 | 140,320 | 137,210 | 134,415 | 131,830 |
| $\mu[EENS]$ (Mwh/y) | $2.36 \times 10^3$ | $5.01 \times 10^3$ | $5.95 \times 10^3$ | $6.69 \times 10^3$ | $8.83 \times 10^3$ |

DTR provides benefits, in order to solve the grid limitations, particularly considering the growing production and consumption of renewable energy resources. When OTS and DTR are coordinated, system efficiency can be improved, in comparison with the SLR scenario. Table 8 shows the exact cost of dispatch and the reliability of the grid criteria for both SLR and DTR case studies, for wind farms with an installation capacity of 20%. As a conclusion, DTR has a lower total system cost compared to SLR. Furthermore, as opposed to the optimal approach, which is to open four lines, the execution of DTR allows the dispatching of the grid generator sets to be less expensive than SLR, where the production cost decreased 2.03% to 6.78%. Figure 9 shows the comparison between DTR and SLR. According to Figure 10, DTR-OTS has the lowest increase in EENS, 7.71% to 8.03%, in comparison with SLR.

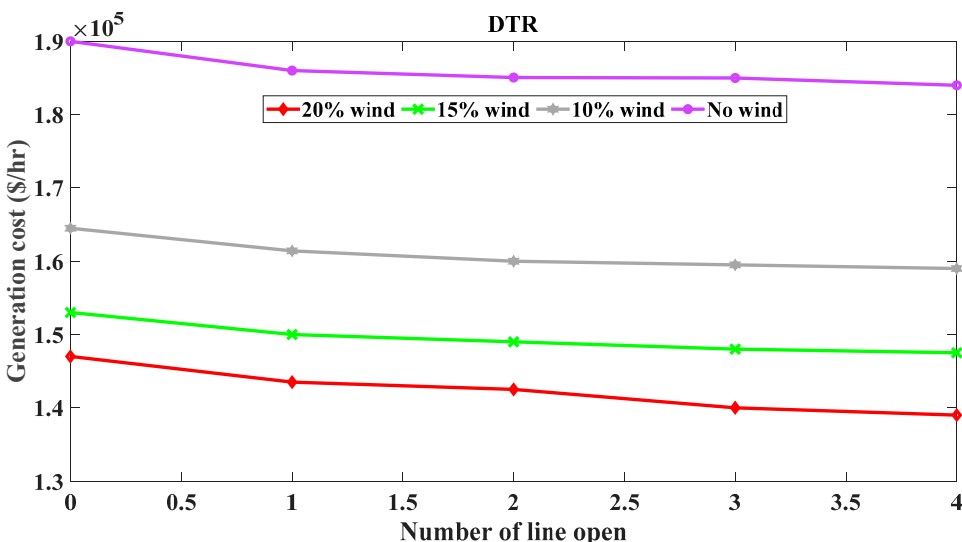

**Figure 7.** Expected producing cost with various DTR wind stages.

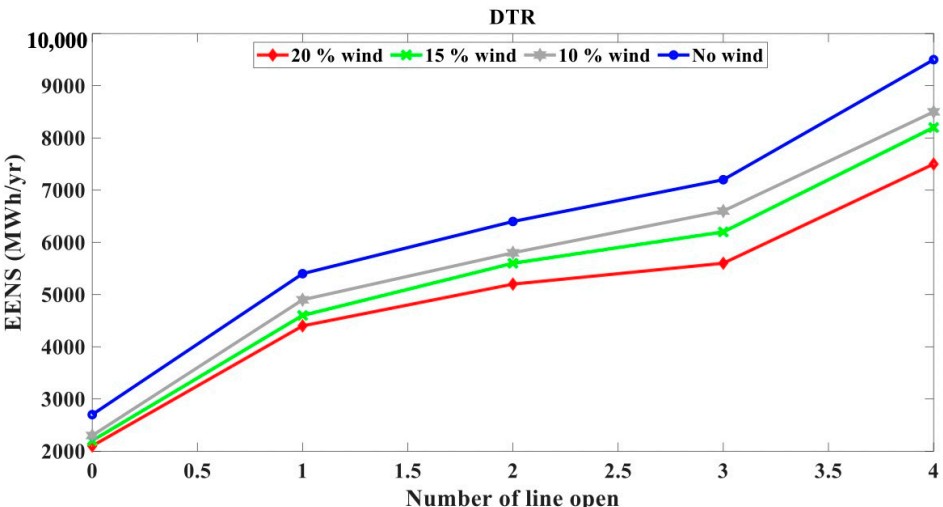

**Figure 8.** EENS for the OTS methods, for various DTR wind stages.

**Table 8.** Comparisons of BCS for OTS methods with DTR and SLR (with wind influence 20%).

| $\varphi$ | 0 | 1 | 2 | 3 | 4 |
|---|---|---|---|---|---|
| SLR Lines open | - | [109–111] | [210,211]; [114–111] | [118–121]; [109–108]; [210–205] | [310–305]; [106–110]; [118–117]; [109–111] |
| $\mu[Gen.cost]$ ($/h) | 146,820 | 143,620 | 142,510 | 140,715 | 140,030 |
| $\mu[EENS]$ (Mwh/y) | $2.41 \times 10^3$ | $5.01 \times 10^3$ | $6.12 \times 10^3$ | $6.77 \times 10^3$ | $8.93 \times 10^3$ |
| DTR Lines open | - | [109–111] | [210,211]; [109–111] | [118–121]; [109–111]; [210,211] | [310,311]; [106–110]; [118–121]; [109–111] |
| $\mu[Gen.cost]$ ($/h) | 141,520 | 140,320 | 137,210 | 134,415 | 131,830 |
| $\mu[EENS]$ (Mwh/y) | $2.10 \times 10^3$ | $4.41 \times 10^3$ | $5.23 \times 10^3$ | $5.64 \times 10^3$ | $7.53 \times 10^3$ |

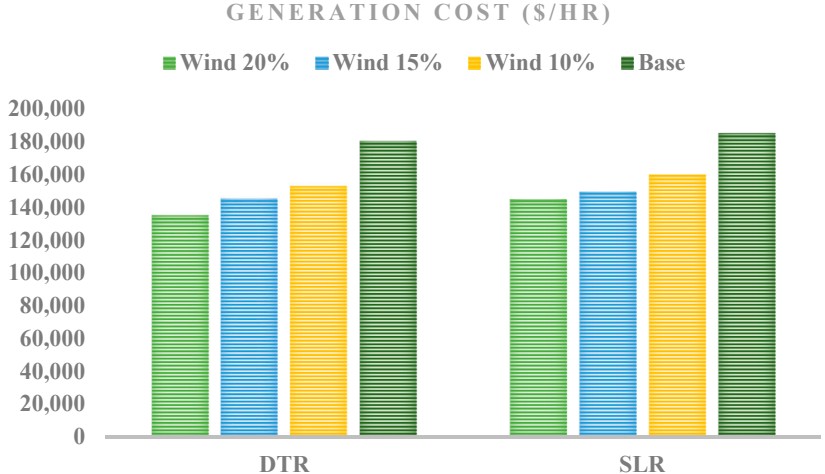

**Figure 9.** Comparing the production costs of DTR and SLR at various wind influence stages.

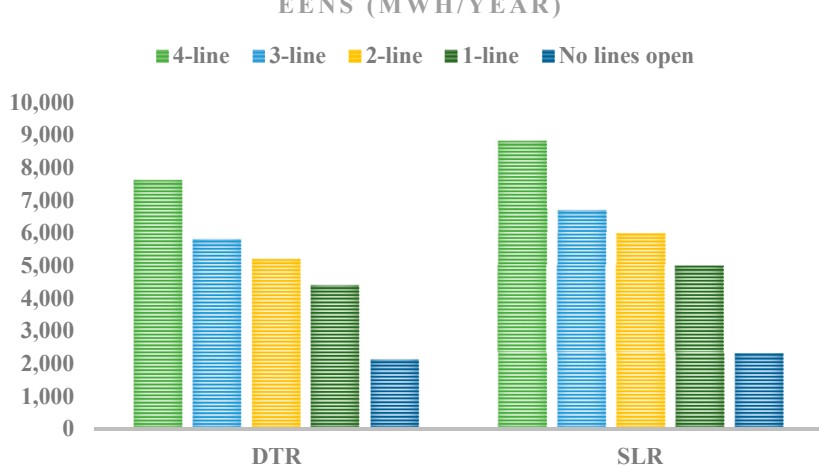

**Figure 10.** Comparing EENS for SLR and DTR for various OTS methods, with wind penetration of 20%.

## 7. Conclusions

This paper uses TS capacity as the primary source for flexibility. The technologies employed in this paper are DTR and OTS, which have both been anticipated to contribute to improved congestion mitigation and enhanced network utilization. As a result, the paper proposes a simple and efficient method to solve the CM problem, by applying the OTS approach, which considers the total cost of production as well as the system reliability index, as multiple objectives with the greatest influence rate on wind production. A modified IEEE RTS-96 system was used to evaluate the validity of the suggested method, and (MOMPSO + PEM) was used to solve it. Based on the simulation outcomes, applying the OTS approach improves the performance of the TS, by regarding the feasibility of altering its topology. Through integration of DTR, the system can achieve a number of benefits, including increased grid capacity. Due to simplified implementation procedures and a lack of major infrastructure or line rebuilding, the DTR method is remarkably appealing in systems that are, presently, suffering from a crescent status of overload. According to the outcomes, OTS could make better use of transmission capacity by using NTO, and DTR can improve reliability and resilience by improving grid operators' awareness of individual assets, thus enhancing flexibility.

**Author Contributions:** Data curation, W.H. and R.Y.M.L.; formal analysis, T.S.; investigation, W.H.; methodology, W.H. and T.S.; project administration, T.S.; software, R.Y.M.L.; supervision, W.H.; validation, R.Y.M.L. All authors have read and agreed to the published version of the manuscript.

**Funding:** This research received no external funding.

**Conflicts of Interest:** The authors declare no conflict of interest.

## Abbreviation

| | |
|---|---|
| ACOPF | AC optimal power flow |
| CM | Congestion management |
| EENS | Expected energy not supplied |
| FACTs | Flexible AC-TS |
| LD | Load demand |
| MPSO | Modified particle swarm optimization |
| MOMPSO | Multi-objective MPSO |
| MCS | Monte Carlo simulation |
| NTO | Network topology optimization |
| OTS | Optimal transmission switching |
| PEM | points estimation method |
| PPF | Probabilistic power flow |
| PDFs | probability distribution functions |
| RO | Robust optimization |
| SOTS | Stochastic optimal transmission switching |
| SLRs | Static line ratings |
| SLR | Static line ratings |
| TL | Transmission lines |
| TN | Transmission network |
| WT | Wind turbine |

## Appendix A

This section provides all the input data used for modeling the uncertainty effects due to the renewable sources and load demand. There are 190 WTs in each wind farm with the specifications as follows:

NEG Micon 1500/64 WT,

Scale parameter c = 8.549 m/s;
Shape parameter k = 1.98;
$V_i$ = 5 m/s;
$V_r$ = 15 m/s;
$V_o$ = 25 m/s;
and $P_r$ = 1.5 MW.

Regarding the PDF functions, all load buses are modeled with the normal density function of the mean value of the base value (active power/reactive power value) and a 7% standard deviation of the base value. For the wind turbine, it is modeled by the Weibull distribution function, with the scale parameter of the base value and the shape parameter of 5% of the base value. It is clear that any other appropriate PDF might be applied, based on the real dataset.

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
