# Peer review of "Management Optimization of Electricity System with Sustainability Enhancement"

_sustainability, doi:10.3390/su14116650_

Round 1
Reviewer 1 Report
The formatting of the References section is inconsistent with regard to initials or full names and the order thereof. Further, some authors have been omitted; Ref 15 omits Robert Kluge, Paula Traynor, and Cody Davis for example.
Recommend a complete review of the Refences section.

Author Response
The formatting of the References section is inconsistent with regard to initials or full names and the order thereof. Further, some authors have been omitted; Ref 15 omits Robert Kluge, Paula Traynor, and Cody Davis for example.
Recommend a complete review of the Refences section
Answer 1: We would like to thank the respected reviewer for his/her effort in reviewing the paper. The authors hope that the respected reviewer will find the responses satisfactory. As per your comment, the entire reference list is double checked and corrected in the revised version.

Reviewer 2 Report
List of abbreviation is missing, example DTR, etc....
It is not clear how the uncertainties associated with wind turbine's power production and load demand (LD) is handled.
Points estimation economic method (PEM), how this differ from traditional ED to dispatch the power generation versus load demand.
Optimal transmission switching (OTS) strategies, how this feasible and does effect with TSO contracts?
Two levels are planned for the complete problem.During the first level, MPSO 291 determines the optimal TL switching strategies from various feasible ones, and the 292 results are transferred to the 2?ℎ level. The 2?ℎ level involves the use of the (2PEM+1) for the solution of the issues of the PPF that can be necessary for determining the 294 production cost and reliability index for different strategies.
How do you relate the 2-level optimization problem to the physical power systems? Do you propose new architecture for the power industry to accomodate the 2-level optimization problem?
What will happen, if the network have other type of renewables expect wind or combination of wind, PV and battery?
Author Response
List of abbreviation is missing, example DTR, etc....
Answer 1: We would like to thank the respected reviewer for his/her effort in reviewing the paper. The authors hope that the respected reviewer will find the responses satisfactory. The abbreviation list is added to the paper which is highlighted in yellow in the new manuscript.
|
ACOPF |
AC optimal power flow |
|
CM |
Congestion management |
|
EENS |
Expected energy not supplied |
|
FACTs |
Flexible AC-TS |
|
LD |
Load demand |
|
MPSO |
Modified particle swarm optimization |
|
MOMPSO |
Multi-objective MPSO |
|
MCS |
Monte Carlo simulation |
|
NTO |
Network topology optimization |
|
OTS |
Optimal transmission switching |
|
PEM |
points estimation method |
|
PPF |
Probabilistic power flow |
|
PDFs |
probability distribution functions |
|
RO |
Robust optimization |
|
SOTS |
Stochastic optimal transmission switching |
|
SLRs |
Static line ratings |
|
SLR |
Static line ratings |
|
TL |
Transmission lines |
|
TN |
Transmission network |
|
WT |
Wind turbine |
---------------------------------------------------------------------------------------------------------------------
It is not clear how the uncertainties associated with wind turbine's power production and load demand (LD) is handled.
Answer 2: Thank you for the comment. This paper has proposed a stochastic framework based on PEM to handle the uncertainties of the problem including the renewable sources output power uncertainties and the load demand uncertainties. Therefore, section 2 is devoted to describes this section completely. The core idea in PEM is to replace the PDF functions with some appropriate fitting concentration points. In this approach, the PEM solves probabilistic problems via a deterministic process, although it needs less computation [13]. Therefore, the PEM would decompose the stochastic problem into 2m+1 equivalent deterministic problems with different probabilities. PEMs have the advantage of requiring basic information about random variables in order to model them effectively. The skewness, variance, and kurtosis of the variables are included in the data. This study uses the (2m+1) scheme, in which the kurtosis of the input random parameters is taken into account, and just one more computation has been performed [26]. More details including the complete formulations are provided in paper which are highlighted in yellow.
The above explanations are added to the paper which are highlighted in yellow.
---------------------------------------------------------------------------------------------------------------------
Points estimation economic method (PEM), how this differ from traditional ED to dispatch the power generation versus load demand.
Answer 3: The main difference between the traditional ED and the proposed PEM based model roots in renewable based power generation and also the multi-objective structure of the proposed model. Therefore, the PEM would break down the single stochastic ED problem into 2m+1 equivalent deterministic framework with different probabilities. It is worth noting that m shows the number of uncertainties in the problem. Therefore, the proposed problem is well equipped by the stochastic structure for modeling the uncertainties of the problem.
---------------------------------------------------------------------------------------------------------------------
Optimal transmission switching (OTS) strategies, how this feasible and does effect with TSO contracts?
Answer 4: OTS strategy can affectively change the structure of the grid and thus provide new power flow paths through which the optimal values of the objective functions would enhance. OTS would then minimize the power losses and cost and would enhance the EENS in the system. From the market point of view, this is very useful and can be considered as a powerful ancillary service for the market players.
The above explanations are added to the paper which are highlighted in yellow in the new manuscript.
---------------------------------------------------------------------------------------------------------------------
Two levels are planned for the complete problem.During the first level, MPSO determines the optimal TL switching strategies from various feasible ones, and the results are transferred to the 2?ℎ level. The 2?ℎ level involves the use of the (2PEM+1) for the solution of the issues of the PPF that can be necessary for determining the 294 production cost and reliability index for different strategies. How do you relate the 2-level optimization problem to the physical power systems? Do you propose new architecture for the power industry to accomodate the 2-level optimization problem?
Answer 6: Please note it that no new structure or architecture is provided for the system in the second layer. In fact, the optimal solution of the first layer contains some sort of uncertainty rooting either from the load demand or renewable sources. In this step, the uncertainty effect should be modeled by breaking down the core problem into 2m+1 deterministic problems with different probabilities. It is worth noting that a two-layer structure is required due to the discrete nature of the problem which necessitates the use of uncertainty modeling after finding the optimal solution in the first layer.
---------------------------------------------------------------------------------------------------------------------
What will happen, if the network have other type of renewables expect wind or combination of wind, PV and battery?
Answer 7: Thank you for the comment. There is no difference in the problem structure and it is generic in this case. It means, adding any new uncertainty source such as a new renewable source is only a new uncertain parameter and is modeled by the use of the PEM. It is clear that increasing the number of uncertainties from m to m+1 would result in 2(m+1)+1 deterministic structures which means 2 new frameworks are generated for handling the new uncertainty source.

Reviewer 3 Report
I have concerns regarding the use of distributions in equations 1, 2, and 4. By the way, equation 1 represents normal distribution, which should be mentioned. I must see goodness of fit test results and plots in order to understand if the distributional assumptions reflect data or not.
Why is the cost function quadratic?
Please create the list of abbreviations and the list of symbols with explanation/description.
Remove extra comma in line 153.
Replace in line 120 “Subjecting to:” to “subject to:”
Author Response
I have concerns regarding the use of distributions in equations 1, 2, and 4. By the way, equation 1 represents normal distribution, which should be mentioned. I must see goodness of fit test results and plots in order to understand if the distributional assumptions reflect data or not.
We would like to thank the respected reviewer for his/her effort in reviewing the paper. The authors hope that the respected reviewer will find the responses satisfactory. Regarding the distribution types, please note it that it is widely accepted in the literature [1] and [24] that normal distribution and Weibull type can be used for modeling the behavior of the load and wind speed/power. Nevertheless, it should be noted that any other PDF function can be used in the same manner without loss of generalization. In other words, we can use other PDF types in the quite same way as we used normal or Weibull functions. Meanwhile since these assumptions are made based on long term big data, there is no way to check their accuracy in this work. In fact, in the practice, we need to first make an initial data analysis to find the most fitting PDF based on our real-time data, and then start working with our model.
The above explanations are added to the paper which are highlighted in yellow in the new manuscript.
---------------------------------------------------------------------------------------------------------------------
Why is the cost function quadratic?
Answer 1: Please note it that it is well perceived that the cost function is considered in a quadratic format since the case study is the transmission system. From a technical point of view, this models the nonlinear opening and closing process of the steam valves which is looks like sinusoidal curves. It is clear that we have to use a linear equation (rather than a quadratic function) if the case study is distribution system.
The above explanations are added to the paper which are highlighted in yellow in the new manuscript.
---------------------------------------------------------------------------------------------------------------------
Please create the list of abbreviations and the list of symbols with explanation/description.
Answer 2: As suggested, the list of abbreviations with detailed explanations are added to the paper which are highlighted in yellow as below:
|
ACOPF |
AC optimal power flow |
|
CM |
Congestion management |
|
EENS |
Expected energy not supplied |
|
FACTs |
Flexible AC-TS |
|
LD |
Load demand |
|
MPSO |
Modified particle swarm optimization |
|
MOMPSO |
Multi-objective MPSO |
|
MCS |
Monte Carlo simulation |
|
NTO |
Network topology optimization |
|
OTS |
Optimal transmission switching |
|
PEM |
points estimation method |
|
PPF |
Probabilistic power flow |
|
PDFs |
probability distribution functions |
|
RO |
Robust optimization |
|
SOTS |
Stochastic optimal transmission switching |
|
SLRs |
Static line ratings |
|
SLR |
Static line ratings |
|
TL |
Transmission lines |
|
TN |
Transmission network |
|
WT |
Wind turbine |
Regarding the symbols, all variables and parameters are explained in the paper body due to the diverse structure of the formulations and the very long definitions which exist for some of the variables.
---------------------------------------------------------------------------------------------------------------------
Remove extra comma in line 153.
Answer 3: Thank you. This is implemented in the revised paper in paper.
---------------------------------------------------------------------------------------------------------------------
Replace in line 120 “Subjecting to:” to “subject to:”
Answer 4: Thank you. This is implemented in the revised paper.
---------------------------------------------------------------------------------------------------------------------

Round 2
Reviewer 3 Report
The data are not described. As a result, many things remain unclear.
Author Response
|
The revised paper is a pleasant reading. All the previous concerns have been addressed. However, it is recommended that authors add all the input data. Answer 1: We would like to thank the respected editor as well as reviewers for their effort in reviewing the paper. The authors hope that the respected editor and reviewers will find the responses satisfactory. In response to your response, an appendix is added to the paper to present all the input data as follows: 8. Appendix This section provides all the input data used for modeling the uncertainty effects due to the renewable sources and load demand. There are 190 WTs in each wind farm with the specifications as follows: NEG Micon 1500/64 WT, Scale parameterc = 8.549 m/s; Shape parameter k = 1.98; V_i=5 m/sec, V_r=15 m/s, V_o=25 m/s and P_r=1.5 MW. Regarding the PDF functions, all load buses are modeled with normal density function of the mean value of base value (active power/reactive power value) and 7% standard deviation of the base value. For the wind turbine, it is modeled by the weibull distribution function with the scale parameter of base value and shape parameter of 5% of the base value. It is clear that any other appropriate PDF might be applied based on the real dataset. --------------------------------------------------------------------------------------------------------------------------- |
